

**Title :**
**ORCHIDEE MICT-LEAK (r5459), a global model for the production, transport and**
**transformation of dissolved organic carbon from Arctic permafrost regions, Part**
**1: Rationale, model description and simulation protocol.**
**Authors:**
**Simon P.K. Bowring[1], Ronny Lauerwald[2], Bertrand Guenet[1], Dan Zhu[1], Matthieu**
**Guimberteau[1,3], Ardalan Tootchi[3], Agnès Ducharne[3], Philippe Ciais[1]**
**Affiliations:**
[1] Laboratoire des Sciences du Climat et de l'Environnement, LSCE, CEA, CNRS, UVSQ,
91191 Gif Sur Yvette, France
[2] Department of Geoscience, Environment & Society, Université Libre de Bruxelles,
1050 Bruxelles, Belgium
[3] Sorbonne Université, CNRS, EPHE, Milieux environnementaux, transferts et
interaction dans les hydrosystèmes et les sols, Metis, 75005 Paris, France
**Abstract**
Few Earth System models adequately represent the unique permafrost soil
biogeochemistry and its respective processes; this significantly contributes to
uncertainty in estimating their responses, and that of the planet at large, to warming.
Likewise, the riverine component of what is known as the 'boundless carbon cycle' is
seldom recognized in Earth System modeling. Hydrological mobilization of organic
material from a ~1330–1580 PgC carbon stock to the river network results either in
sedimentary settling or atmospheric 'evasion', processes widely expected to increase
with amplified Arctic climate warming.   Here, the production, transport and
atmospheric release of dissolved organic carbon (DOC) from high-latitude permafrost
soils into inland waters and the ocean is explicitly represented for the first time in the
land surface component (ORCHIDEE) of a CMIP6 global climate model (IPSL). The
model, ORCHIDEE MICT-LEAK, mechanistically represents (a) vegetation and soil
physical processes for high latitude snow, ice and soil phenomena, and (b) the cycling of
DOC and $CO_2$, including atmospheric evasion, along the terrestrial-aquatic continuum
from soils through the river network to the coast, at 0.5° to 2° resolution.  This paper,
the first in a two-part study, presents the rationale for including these processes in a
high latitude specific land surface model, then describes the model with a focus on novel
process implementations, followed by a summary of the model configuration and
simulation protocol.  The results of these simulation runs, conducted for the Lena River
basin, are evaluated against observational data in the second part of this study.
**1 Introduction**
High-latitude permafrost soils contain large stores of frozen, often ancient and relatively
reactive carbon up to depths of over 30m. Soil warming caused by contemporary
anthropogenic climate change can be expected to destabilize these stores (Schuur et al.,
2015) via microbial or hydrological mobilization following spring/summer thaw and
riverine discharge  (Vonk et al., 2015a) as the permafrost line migrates poleward over
time.  The high latitude soil carbon reservoir may amount to ~1330–1580 PgC (Hugelius
et al., 2013, 2014; Tarnocai et al., 2009) –over double that stored in the contemporary
atmosphere, while the yearly lateral flux of carbon from soils to running waters may



amount to ~40% of net ecosystem carbon exchange (McGuire et al., 2009), the majority
as dissolved organic carbon (DOC).
The fact that, to our knowledge, no existing land surface models are able to adequately
simultaneously respresent this unique high latitude permafrost soil environment, the
transformation of soil organic carbon (SOC) to its eroded particulate and DOC forms and
their subsequent lateral transport, as well as the response of all these to warming,
entails significant additional uncertainty in projecting global-scale biogeochemical
responses to human-induced environmental change.
Fundamental to these efforts is the ability to predict the medium under which carbon
transformation will occur:  in the soil, streams, rivers or sea, and under what
metabolising conditions –since these will determine the process mix that will ultimately
enable either terrestrial redeposition and retention, ocean transfer, or atmospheric
release of permafrost-derived organic carbon.  In the permafrost context, this implies
being able to accurately represent (i) the source, reactivity and transformation of
released organic matter, and; (ii) the dynamic response of hydrological processes to
warming, since water phase determines carbon, heat, and soil moisture availability for
metabolisation and lateral transport.
To this end, we take a specific version of the terrestrial component of the IPSL global
Earth System model (ESM) ORCHIDEE (Organising Carbon and Hydrology In Dynamic
Ecosystems), one that is specifically coded for, calibrated with and evaluated on high
latitude phenomena and permafrost processes, called ORCHIDEE-MICT (Guimberteau et
al., 2018).  This code is then adapted to include DOC production in the soil (ORCHIDEE-
SOM, (Camino-Serrano et al., 2018)),  'priming' of SOC (ORCHIDEE-PRIM,(Guenet et al.,
2016, 2018)) and the riverine transport of DOC and $CO_2$, including in-stream
transformations, carbon and water exchanges with wetland soils and gaseous exchange
between river surfaces and the atmosphere (ORCHILEAK, (Lauerwald et al., 2017)).
The resulting model, dubbed ORCHIDEE MICT-LEAK, herafter referred to as MICT-L for
brevity, is therefore able to represent: (a) Permafrost soil and snow physics,
thermodynamics to a depth of 38m and dynamic soil hydrology to a depth of 2m; (b)
Improved representation of biotic stress response to cold, heat and moisture in high
latitudes; (c) Explicit representation of the active layer and frozen-soil hydrologic
barriers; buildup of soil carbon stocks via primary production and vertical translocation
(turbation) of SOC and DOC; (d) DOC leaching from tree canopies, atmospheric
deposition, litter and soil organic matter, its adsorption/desorption to/from soil
particles, its transport and transformation to dissolved $CO_2$ ($CO^*_{2(aq.)}$) and atmospheric
release, as well as the production and hydrological transport of plant root-zone derived
dissolved $CO_2$; (e) Improved representation of C cycling on floodplains; (f) Priming of
organic matter in the soil column and subsequent decomposition dynamics.   In
combination, these model properties allow us to explore the possibility of reproducing
important emergent phenomena observed in recent empirical studies (Fig. 1) arising
from the interaction of a broad combination of different processes and factors.
To our knowledge very few attempts have been made at the global scale of modelling
DOC production and lateral transfer from the permafrost region that explicitly accounts
for such a broad range of high latitude–specific processes, which in turn allows us to



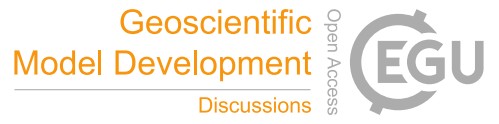

match and evaluate simulation outputs with specific observed processes, enhancing our ability to interpret the output from theses models and improve our understanding of the processes represented. The only other attempt at doing so is a Pan-Arctic modelling study by Kicklighter et al. (2013), which is based on a relatively simplified scheme for soil, water and biology. The following segment briefly overviews the dynamics, emergent properties and their overall significance across scales, of permafrost region river basins.

### *A giant, reactive, fast-draining funnel: A permafrost basin overview*

Permafrost has a profound impact on Arctic river hydrology. In permafrost regions, a permanently frozen soil layer acts as a 'cap' on ground water flow (see 'permafrost barrier', right hand side of Fig. 1). This implies that: (i) Near-surface runoff becomes by far the dominant flowpath draining permafrost watersheds (Ye et al., 2009), as shown in Fig. 1d; (ii) The seasonal amplitude of river discharge, expressed by the ratio of maximum to minimum discharge ($Q_{max:min}$ in Fig. 1), over continuous versus discontinuous permafrost catchments is higher as a result of the permafrost barrier; (iii) This concentration of water volume near the surface causes intense leaching of DOC from litter and relevant unfrozen soil layers (Fig. 1g, 1d, e.g. Drake et al., (2015); Spencer et al., (2015); (Vonk et al., (2015a,b)); (iv) Permafrost SOC stocks beneath the active layer are physically and thermally shielded from aquatic mobilization and metabolization, respectively (Fig. 1g).

Rapid melting of snow and soil or river ice during spring freshet (May-June) drives intensely seasonal discharge, with peaks often two orders of magnitude (e.g. Van Vliet et al., (2012)) above baseflow rates (Fig. 1d). These events are the cause of four, largely synchronous processes: (i) Biogenic matter is rapidly transported from elevated headwater catchments (Fig. 1, right hand side) (McClelland et al., 2016); (ii) Plant material at the soil surface is intensely leached, with subsequent mobilization and transformation of this dissolved matter via inland waters (Fig. 1d,b,j); During spring freshet, riverine DOC concentrations increase and bulk annual marine DOC exports are dominated by the terrestrial DOC flux to the rivers that occurs at this time (Holmes et al., 2012). Indeed, DOC concentrations during the thawing season tend to be greater than or equal to those in the Amazon particularly in the flatter Eurasian rivers (Holmes et al., 2012; McClelland et al., 2012), and DOC concentrations are affected at watershed scale by parent material and ground ice content (O'Donnell et al., 2016).

(iii) Sudden inundation of the floodplain regions in spring and early summer (Fig. 1h), (Smith and Pavelsky, 2008), further spurs lateral flux of both particulate and dissolved matter in the process and its re-deposition (Zubrzycki et al., 2013) or atmospheric evasion (Fig. 1j,m); (iv) Snowmelt-induced soil water saturation, favoring the growth of moss and sedge-based ecosystems (e.g. Selvam et al., 2017; Tarnocai et al., 2009; Yu, 2011) and the retention of their organic matter (OM),.i.e., peat formation, not shown in Fig. 1 as this isn't represented in this model version, but is generated in a separate branch of ORCHIDEE (Qiu et al., 2018)).

Mid-summer river low-flow and a deeper active layer allow for the hydrological intrusion and leaching of older soil horizons (e.g. the top part of Pleistocene-era Yedoma soils), and their subsequent dissolved transport (e.g. Wickland et al., 2018). These





sometimes-ancient low molecular weight carbon compounds appear to be preferentially
and rapidly metabolized by microbes in headwater streams (Fig. 1j), which may
constitute a significant fraction of aggregate summer $CO_2$ evasion in Arctic rivers
(Denfeld et al., 2013; Vonk et al., 2015). This is likely due to the existence of a significant
labile component of frozen carbon (Drake et al., 2015; Vonk et al., 2013; Woods et al.,
153  2011);

$CO_2$ evasion rates from Arctic inland waters (Fig. 1j,e,m) are estimated to be in the
region of 40-84 TgC yr$^{-1}$ (McGuire et al., 2009), to be compared with estimates of Pan
Arctic DOC discharge from rivers of 25-36 TgC yr$^{-1}$.  The influx of terrestrial carbon to
the shelf zone is thought to total 45-54 TgC yr$^{-1}$ (Holmes et al., 2012; Raymond et al.,
2007). Rivers supply the Arctic Ocean (AO) an estimated 34 Tg DOC-C yr$^{-1}$ (Holmes et al.,
2012), while depositing 5.8 Tg yr$^{-1}$ of particulate carbon, these being sourced from those
rivers draining low and high elevation headwaters, respectively (McClelland et al.,
2016).  These dynamics are all subject to considerable amplification by changes in
temperature and hydrology (e.g. Frey et al., 2009; Drake et al., 2015;  Tank et al., 2018).
Average annual discharge in the Eurasian Arctic rivers has increased by at least 7%
between 1936-1999 (Peterson et al., 2002), driven by increasing temperatures and
runoff (Berezovskaya et al., 2005), and the subsequent interplay of increasing annual
precipitation, decreasing snow depth and snow water equivalent (SWE) mass (Kunkel et
al., 2016; Mudryk et al., 2015), and greater evapotranspiration (Zhang et al., 2009).
Although net discharge trend rates over N. America were negative over the period 1964-
2003, since 2003 they have been positive on average (Dery et al., 2016).  These dynamic
and largely increasing hydrologic flux trends point towards temperature and
precipitation -driven changes in the soil column, in which increased soil water/snow
thaw and microbial activity (Graham et al., 2012; MacKelprang et al., 2011; Schuur et al.,
2009) converge to raise soil leaching and DOC export rates to the river basin and beyond
(e.g. Vonk et al., (2015b)). Further, microbial activity generates its own heat, which
incubation experiments have shown may be sufficient to significantly warm the soil
further (Hollesen et al., 2015), in a positive feedback.
Arctic region fire events are also on the rise and likely to increase with temperature and
severity over time (Ponomarev et al., 2016). The initial burning of biomass is
accompanied by active layer deepening, priming of deeper soil horizons (De Baets et al.,
2016), and a significant loading of pyrogenic DOC in Arctic watersheds, up to half of
which is rapidly metabolized (Myers-Pigg et al., 2015).
In these contexts, the implications of (polar-amplified) warmer temperatures leading to
active layer deepening towards the future (transition from Continuous to Discontinuous
Permafrost, as shown in the upper/lower segments of Fig. 1) are clear and unique:
potentially sizeable aquatic mobilization and microbial metabolization (Xue, 2017) of
dissolved and eroded OM, deeper hydrological flow paths, an increase in total carbon
and water mass and heat transfer to the aquatic network and, ultimately, the Arctic
Ocean and atmosphere (Fig. 1i).
The advantage of having a terrestrial model that can be coupled to a marine component
of an overarching global climate model (GCM) is in this case the representation of a
consistent transboundary scheme, such that output from one model is integrated as





input to another. This is particularly important given the context in which these
terrestrial outflows occur :
Because of its small size, a uniquely large and shallow continental shelf, the global
climatological significance of its seasonal sea ice (Rhein et al., 2013) and its rapid decline
(Findlay et al., 2015), the AO has been described as a giant estuary (McClelland et al.,
2012), acting as a funnel for the transport, processing and sedimentation of terrestrial
OM. Because of its small surface area and shallow seas (Jakobsson, 2002), the AO holds
relatively little volume and is consequently sensitive to inputs of freshwater, heat,
alkalinity and nutrients that flush out from terrestrial sources, particularly at discharge
peak.
High suspended particle loads in river water as they approach the mouth (Heim et al.,
2014) cause lower light availability and water albedo and hence higher temperatures
(Bauch et al., 2013; Janout et al., 2016), which can affect the near-shore sea ice extent,
particularly in spring (Steele and Ermold, 2015). Volumes of riverine freshwater and
total energy flux (Lammers et al., 2007) are expected to increase with warmer
temperatures, along with an earlier discharge peak (Van Vliet et al., 2012, 2013). In
doing so, freshwaters may in the future trigger earlier onset of ice retreat (Stroeve et al.,
2014; Whitefield et al., 2015) via a freshwater albedo, ice melt, seawater albedo, ice
melt, feedback, amplified by intermediary state variables such as water vapor and
cloudiness (Serreze and Barry, 2011).
Both terrestrially-exported and older shelf carbon in the AO face considerable
disruption (McGuire et al., 2009; Schuur et al., 2015) from the combined effects of
increased freshwater, heat, sediment, nutrient and organic carbon flows from rapidly
warming Arctic river watersheds, as well as those from melting sea ice, warmer marine
water temperatures and geothermal heat sources (Janout et al., 2016; Shakhova et al.,
2015). Because ORCHIDEE is a sub-component of the overarching IPSL ESM, there is
scope for coupling riverine outputs of water, DOC, $CO_{2(aq)}$ and heat from the terrestrial
model as input for the IPSL marine components (Fig. 1i). Nonetheless, these are not the
objectives of the present paper, whose aim is rather to validate the simulated variable
output produced by the model described in detail below against observations and
empirical knowledge for the Lena basin, but are included here descriptively to scope the
plausible future applications of ORCHIDEE MICT-LEAK, given our present empirical
understanding of their potential significance.
The Methods section summarises the model structure and associated rationale for each
of the model sub-branches or routines relevant to this study, and follows with the setup
and rationale for the simulations carried out as validation exercises.

**2 Methods**
This section overviews the processes represented in the model being described in this
manuscript, which is referred to as ORCHIDEE MICT-LEAK, hereafter referred to MICT-L
for brevity. MICT-L is at its heart a merge of two distinct models : the high-latitude land
surface component of the IPSL Earth System Model ORCHIDEE MICT, and the DOC-
production and transport branch of ORCHIDEE's default or 'trunk' version (Krinner et
al., 2005), ORCHILEAK. The original merger of these two code sets was between

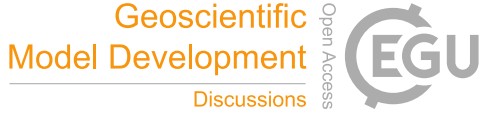

ORCHILEAK and ORCHIDEE-MICT, which are described in Camino-Serrano et al. (2018)/Lauerwald et al. (2017) and Guimberteau et al. (2018), respectively.

However, numerous bug fixes and process additions post-dating these publications have been included in this code. Furthermore, novel processes included in neither of these two core models are added to MICT-L in response to phenomena reported in recent empirical publications, such as the diffusion of DOC (novel in ORCHIDEE-MICT) through the soil column to represent its turbation and preferential stabilisation at depth in the soil, in a process not necessarily the same as its adsorption –also represented here.

In terms of code architecture, the resulting model is substantially different from either of its parents, owing to the fact that the two models were developed on the basis of ORCHIDEE trunk revisions 2728 and 3976 for ORCHILEAK and MICT respectively, which have a temporal model development distance of over 2 years, and subsequently evolved in their own directions. These foundational differences, which mostly affect the formulation of soil, carbon and hydrology schemes, mean that different aspects of each are necessarily forced into the subsequent code. Where these differences were considered scientific or code improvements, they were included in the resulting scheme.

Where these differences were so large as to prove a burden in excess of the scope of this first model version, such as the inclusion of the soil carbon spinup module, they were omitted from this first revision of MICT-L. The direction of the merge –which model was the base which incorporated code from the other –was from ORCHILEAK into MICT, given that the latter contains the bulk of the fundamental (high latitude) processes necessary for this merge. Despite architectural novelties introduced, MICT-L carries with it a marriage of much the same schemes detailed exhaustively in Guimberteau et al. (2018) and Lauerwald et al. (2017). As such, the following model description details only new elements of the model, those that are critical to the production and transport of DOC from permafrost regions, and parameterisations specific to this study (Fig. 2).

## 2.1 Model Description

MICT-L is based largely on ORCHIDEE-MICT, into which the DOC production, transport and transformation processes developed in the ORCHILEAK model version and tested insofar only for the Amazon, have been transplanted, allowing for these same processes to be generated in high latitude regions with permafrost soils and a river flow regime dominated by snow melt. The description that ensues roughly follows the order of the carbon and water flow chain depicted in Fig. 2b. At the heart of the scheme is the vegetative production of carbon, which occurs along a spectrum of 13 plant functional types (PFTs) that differ from one another in terms of plant physiological and phenological uptake and release parameters (Krinner et al., 2005). Together, these determine grid-scale net primary production. In the northern high latitudes, the boreal trees (PFTs 7-9) and C3 grasses (PFT 10) dominate landscape biomass and primary production. Thus, in descending order yearly primary production over the Lena basin is roughly broken down between C3 grasses (48%), boreal needleleaf summergreen trees (27%), boreal needleleaf evergreen trees (12%), boreal broadleaf summergreen trees (8%) and temperate broad-leaved evergreen trees (6%). Naturally these basin aggregates are heterogeneously distributed along latitude and temperature contours, with grasses/tundra dominanting at the high latitudes and (e.g.) temperate broadleaf



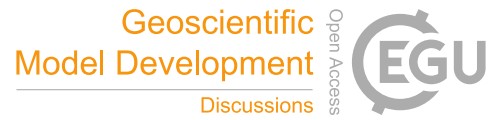

trees existing only at the southern edges of the basin.

**2.2 Biomass generation** (Fig. 1a)

Biomass generation, consisting of foliage, roots, above and below –ground sap and heart
wood, carbon reserves and fruit pools in the model, results in the transfer of these
carbon stores to two downstream litter pools, the structural and metabolic litter (Figure
2b). This distinction, defined by lignin concentration of each biomass pool (Krinner et
al., 2005), separates the relatively reactive litter fraction such as leafy matter from its
less-reactive, recalcitrant counterpart (woody, 'structural' material), with the
consequence that the turnover time of the latter is roughly four-fold that of the former.
These two litterpools are further subdivided into above and below –ground pools, with
the latter explicitly discretised over the first two metres of the soil column, a feature first
introduced to the ORCHIDEE model by Camino-Serrano et al. (2014, 2018). This marks
a significant departure from the original litter formulation in ORCHIDEE-MICT, in which
the vertical distribution of litter influx to the soil carbon pool follows a prescribed root
profile for each PFT. This change now allows for the production of DOC from litter
explicitly at a given soil depth in permafrost soils.

**2.3 DOC generation and leaching** (Fig. 1b)

The vast majority of DOC produced by the model is generated initially from the litter
pools via decomposition, such that half of all of the decomposed litter is returned to the
atmosphere as $CO_2$, as defined by the microbial carbon use efficiency (CUE) –the fraction
of carbon assimilated versus respired by microbes post-consumption –here set at 0.5
following Manzoni et al. (2012). The non-respired half of the litter feeds into 'Active',
'Slow' and 'Passive' free DOC pools, which correspond to DOC reactivity classes in the
soil column. Metabolic litter contributes exclusively to the Active DOC pool, while
Structural litter feeds into the other two, the distribution between them dependent on
the lignin content of the Structural litter. The reactive SOC pools then derive directly
from this DOC reservoir, in that fractions of each DOC pool, defined again by the CUE, are
directly transferred to three different SOC pools, while the remainder adds to the
heterotrophic soil respiration. Depending on clay content and bulk density of the soil, a
fraction of DOC is adsorbed to the mineral soil and does not take part in these reactions
until it is gradually desorbed when concentrations of free DOC decrease in the soil
column. This scheme is explained in detail in Camino-Serrano (2018). The value of the
fractional redistributions between free DOC and SOC after adsorption are shown in Fig.
2b.

The approximate ratio of relative residence times for the three SOC pools in our model
(Active :Slow :Passive) is (1 :37:1618) at a soil temperature of 5°C, or 0.843 years, 31
yrs. and 1364 yrs. for the three pools respectively (Fig. 2b). These are based on our
own exploratory model runs and subsequent calculations. The residence times of the
active DOC pool is ~7 days (0.02 yrs.), while the slow and passive DOC pools both have a
residence time of ~343 days (0.94 yrs.) at that same temperature. Upon microbial
degradation in the model, SOC of each pool reverts either to DOC or to $CO_2$, the ratio
between these determined again by the CUE which is set in this study at 0.5 for all donor
pools, in keeping with the parameter configuration in Lauerwald et al., (2017) from
Manzoni et al. (2012). This step in the chain of flows effectively represents leaching of



SOC to DOC. Note that the reversion of SOC to DOC occurs only along Active-Active,
Slow-Slow and Passive-Passive lines in Fig. 2b, while the conversion of DOC to SOC is
distributed differently so as to build up a reasonable distribution of soil carbon stock
reactivities.   Note also that the microbial CUE is invoked twice in the chain of carbon
breakdown, meaning that the 'effective' CUE of the SOC-litter system is approximately
349   0.25.
**2.4 Throughfall and its DOC** (Fig. 1c)
In MICT-L, DOC generation also occurs in the form of wet and dry atmospheric
deposition and canopy exudation, collectively attributed to the throughfall, i.e. the
amount of precipitation reaching the ground.  Wet atmospheric deposition originates
from organic compounds dispersed in atmospheric moisture which become deposited
within rainfall, and are assumed here to maintain a constant concentration. This
concentration we take from the average of reported rainfall DOC concentrations in the
empirical literature measured at sites >55°N (Bergkvist and Folkeson, 1992; Clarke et
al., 2007; Fröberg et al., 2006; Lindroos et al., 2011; Rosenqvist et al., 2010; Starr et al.,
2003; Wu et al., 2010), whose value is 3 mgC L$^{-1}$ of rainfall. Dry DOC deposition occurs
through aerosol-bound organic compounds, here assumed to fall on the canopy ; canopy
exudation refers to plant sugars exuded from the leaf surface (e.g. honey dew) or from
their extraction by heterotrophs such as aphids.  These two are lumped together in our
estimates of canopy DOC generation (gDOC per g leaf carbon), which is calibrated as
follows.
We take the average total observation-based throughfall DOC flux rate per m$^2$ of forest
from the aforementioned literature bundle (15.7 gC m$^{-2}$ yr$^{-1}$) and subtract from it the
wet deposition component (product of rainfall over our simulation area and the rain
DOC content).  The remainder is then the canopy DOC, which we scale to the average leaf
biomass simulated in a 107-year calibration run over the Lena river basin, to obtain a
constant, non-conservative canopy DOC production rate of 9.2*10$^{-4}$ g DOC-C per gram
leaf biomass per day (Eq. 1), except for the crop PFTs for which this value equals 0. Note
that this production of DOC should be C initially fixed by photosynthesis, but it is here
represented as an additional carbon flux. The dry  deposition of DOC through the canopy
is given by:

$$(1) \quad \textbf{TF}_{\textbf{DRY}} = \textbf{M}_{\textbf{LEAF}} * \textbf{9.2} * \textbf{10}^{-4} \frac{dt}{day}$$

Where TF$_{DRY}$ is dry deposition of DOC from the canopy, M$_{LEAF}$ is leaf biomass, dt is the
timestep of the surface hydrology and energy balance module (30min) and day is 24
hours. This accumulates in the canopy and can be flushed out with the throughfall and
percolates into the soil surface or adds to the DOC stock of surface waters.  The wet and
canopy deposition which hits the soil is then assumed to be split evenly between the
labile and refractory DOC pools (following Aitkenhead-Peterson et al., 2003).
**2.5 Hydrological mobilisation of soil DOC** (Fig. 1d)





All DOC pools, leached from the decomposition of either litter and SOC or being
throughfall inputs, reside at this point in discrete layers within the soil column, but are
now also available for vertical advection and diffusion,  as well as lateral export from the
soil column as a carbon tracer, via soil drainage and runoff.
Export of DOC from the soil to rivers occurs through surface runoff, soil-bottom
drainage, or flooding events (see sections 'soil flooding' and 'floodplain representation').
Runoff is activated when the maximum water infiltration rate of the specific soil has
been exceeded, meaning that water arrives at the soil surface faster than it can enter,
forcing it to be transported laterally across the surface.   DOC is drawn up into this
runoff water flux from the first 5 layers of the soil column, which correspond to a
cumulative source depth of 4.5cm.
Drainage of DOC occurs first as its advection between the discrete soil layers, and its
subsequent export from the 11th layer, which represents the bottom of the first 2m of
the soil column, from which export is calculated as a proportion of the DOC
concentration at this layer. Below this, soil moisture and DOC concentrations are no
longer explicitly calculated, except in the case that they are cryoturbated below this, up
to a depth of 3m.  DOC drainage is proportional to but not a constant multiplier of the
water drainage rate for two reasons. First, in the process of drainage DOC is able to
percolate from one layer to another, through the entirety of the soil column, meaning
that vertical transport is not solely determined by 11th layer concentrations, given that
DOC can be continuously leached and transported over the whole soil column.  Secondly,
in order to account for preferential flow paths in the soil created by the subsoil actions
of flora and fauna, and for the existence of non-homogenous soil textures at depth that
act as aquitards, DOC infiltration must account for the fact that area-aggregated soils
drain more slowly, increasing the residence time of DOC in the soil.  Thus a reduction
factor which reduces the vertical advection of DOC in soil solution by 80% compared to
the advection is applied to represent a slow down in DOC percolation through the soil
and increase its residence time there.
In MICT-L, as in ORCHILEAK, a 'poor soils' module reads off from a map giving fractional
coverage of land underlain by Podzols and Arenosols at the 0.5° grid-scale, as derived
from the  Harmonized World Soil Database (Nachtergaele, 2010). Due to their low pH
and nutrient levels, areas identified by this soil-type criterion experience soil organic
matter decomposition rates half that of other soils (Lauerwald et al. (2017), derived
from Bardy et al. (2011); Vitousek & Sanford (1986); Vitousek & Hobbie (2000)). To
account for the very low DOC-filtering capacity of these coarse-grained, base- and clay-
poor soils (DeLuca & Boisvenue (2012), Fig. 2b), no reduction factor in DOC advection
rate relative to that of water in the soil column is applied when DOC is generated within
these "poor soils"..
By regulating both decomposition and soil moisture flux, the "poor soil" criterion
effectively serves a similar if not equal function to a soil 'tile' for DOC infiltration in the
soil column (inset box of Fig. 1), because soil tiles (forest, grassland/tundra/cropland
and bare soil) are determinants of soil hydrology which affects moisture-limited
decomposition.  Here however, the 'poor soil' criteria is applied uniformly across the
three soil tiles of ecah grid cell. This modulation in MICT-L is of significance for the
Arctic region, given that large fractions of the discontinous permafrost region are





underlain by Podzols, particularly in Eurasia. For the Arctic as a whole, Podzols cover
~15% of total surface area (DeLuca and Boisvenue, 2012). Further, in modelled frozen
soils, a sharp decline in hydraulic conductivity is imposed by the physical barrier of ice,
which retards the flow of water to depth in the soil, imposing a cap on drainage and thus
potentially increasing runoff of water laterally, across the soil surface (Gouttevin et al.,
2012). In doing so, frozen soil layers overlain by liquid soil moisture will experience
enhanced residence times of water in the carbon-rich upper soil layers, potentially
enriching their DOC load.
Thus, for all the soil layers in the first 2m, DOC stocks are controlled by production from
litter and SOC decay, their advection, diffusion, and consumption by DOC mineralization,
as well as buffering by adsorption and desorption processes.
**2.6 Routing Scheme:**
The routing scheme in ORCHIDEE, first described in detail in Ngo-Duc et al. (2007) and
presented after some version iterations in Guimberteau et al. (2012), is the module
which when activated, represents the transport of water collected by the runoff and drainage
simulated by the model along the prescribed river network in a given watershed. In doing so,
its purpose is to coarsely represent the hydrologic coupling between precipitation
inputs to the model and subsequent terrestrial runoff and drainage (or evaporation)
calculated by it on the one hand, and the eventual discharge of freshwater to the marine
domain, on the other. In other words, the routing scheme simulates the transport of
water by rivers and streams, by connecting rainfall and continental river discharge with
the land surface.
To do so, the routing scheme first inputs a map of global watersheds at the 0.5 degree
scale (Oki et al., 1999; Vorosmarty et al., 2000) which gives watershed and sub-basin
boundaries and the direction of water-flow based on topography to the model. The
water flows themselves are comprised of three distinct linear reservoirs within each sub-basin
('slow', 'fast', 'stream'). Each water reservoir is represented at the subgrid scale (here: 4
subgrid units per grid cell), and updated with the lateral in- and outflows at a daily time-step.
The 'slow' water reservoir aggregates the soil drainage, i.e. the vertical outflow from the 11[th]
layer (2 m depth) of the soil column, effectively representing the 'shallow groundwater'
storage. The 'fast' water reservoir aggregates surface runoff simulated in the model,
effectively representing overland hydrologic flow. The 'slow' and 'fast' wast reservoirs feed a
delayed outflow to the 'stream' reservoir' of the adjacent subgrid-unit in the downstream
direction.
The water residence time in each reservoir depends on the nature of the reservoir (increasing
residence time in the order : stream < fast < slow reservoir). More generally, residence time
decreases with the steepness of topography, given by the product of a local topographic
index and a constant with decreasing values for the 'slow', 'fast' and 'stream' reservoirs.
The topographic index is the ratio of the grid-cell length to the square root of the mean
slope, to reproduce the effect of geomorphological factors in Manning's equation
(Ducharne et al., 2003; Guimberteau et al., 2012; Manning, 1891) and determines the
time that water and DOC remain in soils prior to entering the river network. In this way
the runoff and drainage are exported from sub-unit to sub-unit and from grid-cell to grid-cell.



**2.7 Grid-scale water and carbon routing** (Fig. 1f, 1g)

Water-borne, terrestrially-derived DOC and dissolved $CO_2$ in the soil solution are exported over the land surface using the same routing scheme. When exported from soil or litter, DOC remains differentiated in the numerical simulations according to its initial reactivity within the soil (Active, Slow, Passive). However, because the terrestrial Slow and Passive DOC pools (Camino-Serrano et al., 2018) are given the same residence time, these two pools are merged when exported (Lauerwald et al., 2017): Active DOC flows into a Labile DOC hydrological export pool, while the Slow and Passive DOC pools flow into a Refractory DOC hydrological pool (Fig. 2b). The water residence times in each reservoir of each subgrid-unit determine the decomposition of DOC into $CO_2$ within water reservoirs, before non-decomposed DOC is passed on to the next reservoir downstream.

The river routing calculations, which occur at a daily timestep, are then aggregated to one-day for the lateral transfer of water, $CO_{2(aq)}$ and DOC from upstream grid to downstream grid according to the river network. Note that carbonate chemistry in rivers and total alkalinity routing are not calculated here.

In this framework, the 'fast' and 'slow' residence times of the water pools in the routing scheme determine the time that water and DOC remains in overland and groundwater flow before entering the river network. Note that while we do not explicitly simulate headwater streams as they exist in a geographically determinant way in the real world, we do simulate what happens to the water before it flows into a river large enough to be represented in the routing scheme by the water pool called 'stream'. The 'fast' reservoir, which is indicative of the pool of runoff water that is destined for entering the 'stream' water reservoir, is implicitly representative of headwater streams non resolved by the model routing as an explicit stream pool at a given spatial resolution, as it fills the spatial and temporal niche between runoff and the river stem. The dynamics of headwater hydrological and DOC dynamics (Section 2.10) are of potentially great significance with respect to carbon processing, as headwater catchments have been shown to be 'hotspots' of carbon metabolisation and outgassing in Arctic rivers, despite their relatively small areal fraction (Denfeld et al., 2013; Drake et al., 2015; Mann et al., 2015; Venkiteswaran et al., 2014; Vonk et al., 2013, 2015a, 2015b). Thus, in what follows in this study, we refer to what in the code are called the 'fast' and 'stream' pools, which represent the small streams and large stream or river pools, respectively, using 'stream' and 'river' to denote these from hereon in.

Furthermore, the differentiated representation of water pools as well as mean grid cell slope, combined with the dynamic active layer simulated for continuous versus discontinuous permafrost, is important for reproducing the phenomena observed by Kutscher et al. (2017) and Zhang et al. (2017) for sloping land as shown on the right hand side of Fig. 1. In discontinuous permafrost and permafrost free regions, these phenomena encompass landscape processes (sub-grid in the model), through which water flow is able to re-infiltrate the soil column and so leach more refractory DOC deeper in the soil column, leading to a more refractory signal in the drainage waters. In contrast, in continuous permafrost region, the shallow active layer will inhibit the downward re-infiltration flux of water and encourage leaching at the more organic-rich and labile surface soil layer, resulting in a more labile DOC signal from the drainage in





these areas (Fig. 1). These re-infiltration processes are thought to be accentuated in
areas with higher topographic relief (Jasechko et al., 2016), which is why they are
represented on sloping areas in Fig. 1.
**2.8 Representation of floodplain hydrology and their DOC budget** (Fig. 1e,1h)
The third terrestrial DOC export pathway in MICT-L is through flooding of floodplains, a
transient period that occurs when stream water is forced by high discharge rates over
the river 'banks' and flows onto a flat floodplain area of the grid cell that the river
crosses, thus inundating the soil. Such a floodplain area is represented as a fraction of a
grid-cell with the maximum extent of inundation, termed the 'potential flooded area'
being predefined from a forcing file (Tootchi et al., 2019). Here, the DOC pools that are
already being produced in these inundated areas from litter and SOC decomposition in
the first 5 layers of the soil column are directly absorbed by the overlying flood waters.
These flood waters may then either process the DOC directly, via oxidisation to $CO_2$,
(Sections 2.10, 2.11) or return them to the river network, as floodwaters recede to the
river main stem, at which point they join the runoff and drainage export flows from
upstream.
MICT-L includes the floodplain hydrology part of the routing scheme (D'Orgeval et al.,
2008; Guimberteau et al., 2012), as well as additions and improvements described in
Lauerwald et al. (2017). The spatial areas that are available for potential flooding are
pre-defined by an input map originally based on the map of Prigent et al. (2007).
However, for this study, we used an alternative map of the "regularly flooded areas"
derived from the method described in Tootchi et al., (2018), which in this study uses an
improved input potential flooding area forcing file specific to the Lena basin, that
combines three high-resolution surface water and inundation datasets derived from
satellite imagery: GIEMS-D15 (Fluet-Chouinard et al., 2015), which results from the
downscaling of the map of Prigent et al. (2007) at 15-arc-sec (ca 500 m at Equator);
ESA-CCI land cover (at 300 m ~ 10 arc-sec); and JRC surface water at 1 arc-sec (Pekel et
al., 2016). The 'fusion' approach followed by this forcing dataset stems from the
assumption that the potential flooding areas identified by the different datasets are all
valid despite their uncertainties, although none of them is exhaustive. The resulting map
was constructed globally at the 15 arc-sec resolution and care was taken to exclude
large permanent lakes from the potential flooding area based on the HydroLAKES
database (Messager et al., 2016). In the Lena river basin, the basin against which we
evaluate ORCHIDEE MICT-LEAK in Part 2 of this study, this new potential floodplains file
gives a maximum floodable area of 12.1% ($2.4*10^5$ km$^2$) of the $2.5*10^6$ km$^2$ basin,
substantially higher than previous estimates of 4.2% by Prigent et al. (2007).
With this improved forcing, river discharge becomes available to flood a specific pre-
defined floodplain grid fraction, creating a temporary floodplains hydrologic reservoir,
whose magnitude is defined by the excess of discharge at that point over a threshold
value, given by the median simulated water storage of water in each grid cell over a 30
year period. The maximum extent of within-grid flooding is given by another threshold,
the calculated height of flood waters beyond which it is assumed that the entire grid is
inundated. This height, which used to be fixed at 2 m, is now determined by the 90th
percentile of all flood water height levels calculated per grid cell from total water
storage of that grid cell over a reference simulation period for the Lena basin, using the



same methodology introduced by Lauerwald et al. (2017).  The residence time of water
on the floodplains ( $\tau_{flood}$) is a determinant of its resulting DOC concentration, since
during this period it appropriates all DOC produced by the top 5 layers of the soil
column.
**2.9 Oceanic outflow** (Fig. 1i)
Routing of water and DOC through the river network ultimately lead to their export
from the terrestrial system at the river mouth (Fig. 1), which for high latitude rivers are
almost entirely sub-deltas of the  greater 'estuary', described by McClelland et al. (2012),
draining into the Arctic Ocean.   Otherwise, the only other loss pathway for carbon
export once in the river network is through its decomposition to $CO_2$ and subsequent
escape to the atmosphere from the river surface.  DOC decomposition is ascribed a
constant fraction for the labile and refractory DOC pools of 0.3 d$^{-1}$ and 0.01 d$^{-1}$ at 25°C,
respectively, these modulated by a water-temperature dependent Arrhenius rate term.
Because the concentration of dissolved $CO_2$ (referred to as $CO_{2(aq.)}$) in river water is
derived not only from in-stream decomposition of DOC, but also from $CO_{2(aq.)}$ inputs
from the decomposition of litter, SOC and DOC both in upland soils and in inundated
soils, the model also represents the lateral transport of $CO_{2(aq.)}$ from soils through the
river network.  Note that autochtonous primary production and derivative carbon
transformations are ignored here, as they are considered relatively minor contributors
in the Arctic lateral flux system (Cauwet and Sidorov, 1996; Sorokin and Sorokin, 1996).
**2.10 Dissolved $CO_2$ export and river evasion** (Fig. 1j)
Soil $CO_{2(aq.)}$ exports are simulated by first assuming a constant concentration of $CO_{2(aq.)}$
with surface runoff and drainage water fluxes, of 20 and 2 mgC L$^{-1}$, corresponding to a
$p$CO2 of 50000 $\mu$ atm and 5000 $\mu$ atm at 25°C in the soil column, respectively.  These
quantities are then scaled with total (root, microbial, litter) soil respiration by a  scaling
factor first employed in Lauerwald et al. (2018, *in review*). In the high latitudes soil
respiration is dominantly controlled by microbial decomposition, and for the Lena basin
initial model tests suggest that its proportional contribution to total respiration is
roughly 90%, versus 10% from root respiration. Thus $CO_{2(aq.)}$ enters and circulates the
rivers via the same routing scheme as that for DOC and river water. The lateral transfers
of carbon are aggregated from the 30 minute time steps at which they are calculated,
with a 48 timestep period, so that they occur within the model as a daily flux. The
calculation of the river network pCO2 can then be made from $CO_{2(aq.)}$ and its equilibrium
with the atmosphere, which is a function of its solubility ($K_{CO2}$) with respect to the
temperature of the water surface $T_{WATER}$ (Eq.2).

$$(2)\ \ pCO2_{POOL} = \frac{[CO_{2(aq)}]}{12.011 * K_{CO_2}}$$

Where the pCO2 of a given (e.g. 'stream', 'fast', 'slow' and floodplain) water pool
($p$CO2$_{POOL}$) is given by [$CO_{2(aq)}$] the dissolved $CO_2$ concentration in that pool, and $K_{CO2}$.
Water temperature ($T_{WATER}$, (°C)) isn't simulated by the model, but is derived here from
the average daily surface temperature ($T_{GROUND}$, (°C)) in the model (Eq. 3), a set up used
by Lauerwald et al. (2017) and retained here. Note that while dissolved $CO_2$ enters from



the terrestrial reservoir from organic matter decomposition, it is also generated *in situ*
within the river network as DOC is respired microbially.
With our water temperature estimate, both $K_{CO2}$ and the Schmidt number (Sc) from
Wanninkhof (1992) can be calculated, allowing for simulation of actual gas exchange
velocites from standard conditions. The Schmidt number links the gas transfer velocity
of any soluble gas (in this case carbon dioxide) from the water surface to water
temperature. For more on the Schmidt number see (Wanninkhof, 2014, 1992). The $CO_2$
that evades is then subtracted from the [$CO_2$] stocks of each of the different hydrologic
reservoirs –river, flood and stream.

$$(3)\quad T_{WATER} = 6.13°C + (0.8 * T_{GROUND})$$

$$(4)\quad Sc = ((1911 - 118.11) * T_{WATER}) + (3.453 * T_{WATER}^2) - (0.0413 * T_{WATER}^3)$$

$CO_2$ evasion is therefore assumed to originate from the interplay of $CO_2$ solubility,
relative gradient in partial pressures of $CO_2$ between air and water, and gas exchange
kinetics. Evasion as a flux from river and floodplain water surfaces is calculated at a
daily timestep, however in order to satisfy the sensitivity of the relative gradient of
partial pressures of $CO_2$ in the water column and atmosphere to both $CO_2$ inputs and
evasion, the *p*$CO_2$ of water is calculated at a more refined 6 minute timestep. The daily
lateral flux of $CO_2$ inputs to the water column are thus equally broken up into 240 (6
min.) segments per day and distributed to the *p*$CO_2$ calculation. Other relevant carbon
processing pathways, such as the photochemical breakdown of riverine OC, are not
explicitly included here, despite the suggestion by some studies that the photochemical
pathway dominate DOC processing in Arctic streams (e.g. Cory et al., 2014). Rather,
these processes are bundled into the aggregate decomposition rates used in the model,
which thus include both microbial and photochemical oxidation. This is largely because
it is unclear how different factors contribute to breaking down DOC in a dynamic
environment and also the extent to which our DOC decomposition and $CO_2$ calculations
implicitly include both pathways –e.g. to what extent the equations and concepts used in
their calculation confound bacterial with photochemical causation, since both microbial
activity and incident UV light are a function of temperature and total incident light.
**2.11 Soil layer processes: turbation** (Fig. 1k), **adsorption** (Fig. 1l)
The soil carbon module is discretised into a 32-layer scheme totalling 38m depth, which
it shares with the soil thermodynamics to calculate temperature through the entire
column. An aboveground snow module (Wang et al., 2013) is discretised into 3 layers of
differing thickness, heat conductance and density, which collectively act as a
thermodynamically-insulating intermediary between soil and atmosphere (Fig. 2a).
Inputs to the three soil carbon pools are resolved only for the top 2m of the soil, where
litter and DOC are exchanged with SOC in decomposition and adsorption/desorption
processes. Decomposition of SOC pools, calculated in each soil layer, is dependent on
soil temperature, moisture and texture (Koven et al., 2009; Zhu et al., 2016), while
vertical transfer of SOC is enabled by representation of cryoturbation (downward
movement of matter due to repeated freeze-thaw) in permafrost regions, and
bioturbation (by soil organisms) in non-permafrost regions in terms of a diffusive flux.



Cryoturbation, given a diffusive mixing rate (Diff) of 0.001 m$^2$ yr$^{-1}$ (Koven et al., 2009), is possible to 3 m depth (diffusive rate declines linearly to zero from active layer bottom to 3 m), and extends the soil column carbon concentration depth in permafrost regions from 2 m. Bioturbation is possible to 2 m depth, with a mixing rate of of 0.0001 m$^2$ yr$^{-1}$ (Koven et al., 2013) declining to zero at 2 m (Eq. 5). In MICT-L, these vertical exchanges in the soil column are improved on. Now, we explicitly include the cryoturbation and bioturbation of both belowground litter and DOC. These were not possible in ORCHIDEE-MICT because, for the former, the belowground litter distribution was not explicitly discretised or vertically dynamic, and for the latter because DOC was not produced in prior versions. Diffusion is given by :

$$((5)\ )\quad \frac{\delta DOC_i(z)}{\delta t} = IN_{DOC_i}(z) - k_i(z) * \phi * DOC_i(z) + Diff\frac{\delta DOC_i^2(z)}{\delta z^2}$$

Where DOC$_i$ is the DOC in pool i at depth z, (gC m$^{-3}$) IN$_{DOCi}$ the inflow of carbon to that pool (gCm$^{-3}$d$^{-1}$), k$_i$ the decomposition rate of that pool (d$^{-1}$), Φ the temperature dependent rate modifier for DOC decomposition and *Diff* the diffusion coefficient (m$^2$ yr$^{-1}$). The vertical diffusion of DOC in non-permafrost soils represented here (that is, the non-cryoturbated component) appears to be consistent with recent studies reporting an increased retention of DOC in the deepening active layer of organic soils (Zhang et al., 2017). This vertical translocation of organic carbon, whether in solid/liquid phase appears to be an important component of the high rates of SOC buildup observed at depth in deep permafrost soils.

**2.11 Priming** (Fig. 1m)

MICT-L also incorporates a scheme for the 'priming' of organic matter decomposition, a process in which the relative stability of SOC is impacted by the intrusion of or contact with SOC of greater reactivity, resulting in enhanced rates of decomposition. This was first introduced by Guenet et al. (2016) and updated in Guenet et al. (2018). This process has shown itself to be of potentially large significance for SOC stocks and their respiration in high latitude regions, in empirical in situ and soil incubation studies (De Baets et al., 2016; Walz et al., 2017; Wild et al., 2014, 2016; Zhang et al., 2017), as well as modelling exercises (Guenet et al., 2018). Here, priming of a given soil pool is represented through the decomposition of soil carbon (dSOC/dt) by the following equation :

$$(6)\quad \frac{dSOC}{dt} = IN_{SOC} - k * (1 - e^{-c*FOC}) * SOC * \Theta * \phi * \gamma$$

Where IN$_{SOC}$ is the carbon input to that pool, k is the SOC decomposition rate, FOC is a stock of matter interacting with this SOC pool to produce priming, c is a parameter controlling this interaction, SOC is the SOC reservoir, and $\theta$, Φ and $\gamma$ the moisture, temperature and texture functions that modulate decomposition in the code. The variable FOC ('fresh organic carbon') is an umbrella term used for specifying all of the carbon pools which together constitute that carbon which is considered potential priming donor material –ie. more labile – to a given receptor carbon pool. Thus, for the slow soil carbon pool FOC incorporates the active soil carbon pool plus the above and below ground structural and metabolic litter pools, because these pools are donors to





the slow pool, and considered to accelerate its turnover through priming. Importantly,
previous studies with priming in ORCHIDEE employed this scheme on a version which
resolves neither the vertical discretisation of the soil column nor the explicit vertical
diffusion processes presented here. This is potentially significant, since the vertical
diffusion of relatively reactive matter may strongly impact (accelerate) the
decomposition of low reactivity matter in the deeper non-frozen horizons of high
latitude soils, while the explicit discretisation of the soil column is a significant
improvement in terms of the accuracy of process-representation within the column
itself.
Other carbon-relevant schemes included in MICT-L are: A prognostic fire routine
(SPITFIRE), calibrated for the trunk version of ORCHIDEE (Yue et al., 2016) is available
in our code but not activated in the simulations conducted here. As a result, we do not
simulate the ~13% of Arctic riverine DOC attributed to biomass burning by Myers-Pigg
et al. (2015), or the ~8% of DOC discharge to the Arctic Ocean from the same source
(Stubbins et al., 2017). Likewise, a crop harvest module consistent with that in
ORCHIDEE-MICT exists in MICT-L but remains deactivated for our simulations.
A module introduced in the last version of ORCHIDEE-MICT (Guimberteau et al., 2018),
in which the soil thermal transfer and porosity and moisture are strongly affected by
SOC concentration, is deactivated here, because it is inconsistent with the new DOC
scheme. Specifically, while carbon is conserved in both MICT and MICT-L soil schemes,
MICT-L introduces a new reservoir into which part of the total organic carbon in the soil
–the DOC –must now go. This then lowers the SOC concentration being read by this
thermix module, causing significant model artefact in soil thermodynamics and
hydrology in early exploratory simulations. Ensuring compatibility of this routine with
the DOC scheme will be a focal point of future developments in MICT-L. Other processes
being developed for ORCHIDEE-MICT, including a high latitude peat formation (Qiu et
al., 2018), methane production and microbial heat generating processes that are being
optimised and calibrated, are further pending additions to this particular branch of the
ORCHIDEE-MICT series.

### 3 Soil Carbon Spinup and Simulation Protocol

The soil carbon spinup component of ORCHIDEE, which is available to both its trunk and
MICT branches, was omitted from this first version of MICT-L, owing to the code burden
required for ensuring compability with the soil carbon scheme in MICT-L. However,
because we are simulating high latitude permafrost regions, having a realistic soil
carbon pool at the outset of the simulations is necessary if we are to untangle the
dynamics of SOC and DOC with a changing environment. Because the soil carbon spinup
in ORCHIDEE-MICT is normally run over more than 10,000 years (Guimberteau et al.,
2108), and because running MICT-L for this simulation period in its normal, non-spinup
simulation mode would impose an unreasonable burden on computing resources, here
we directly force the soil carbon output from a MICT spinup directly into the restart file
of a MICT-L simulation.
A 20,000 year spinup loop over 1961-1990 (these years chosen to mimic coarsely
warmer mid-Holocene climate) -forced by GSWP-3 climatology, whose configuration
derives directly from that used in Guimberteau et al. (2018), was thus used to replace



the three soil carbon pool  values from a 1-year MICT-L simulation to set their initial
values.  A conversion of this soil carbon from volumetric to areal units was applied,
owing to different read/write standards in ORCHILEAK versus ORCHIDEE-MICT. This
artifically imposed, MICT-derived SOC stock would then have to be exposed to MICT-L
code, whose large differences in soil carbon module architecture as compared to MICT,
would drive a search for new equilibrium soil carbon stocks.
Due to the long residence times of the passive SOC pool, reaching full equilibrium for it
requires a simulation length on the order of 20,000y –again an overburden. As we are
interested primarily in DOC in this study, which derives mostly from the Active and Slow
SOC pools, the model was run until these two pools reached a quasi-steady state
equilibria (Part 2 Supplement, Fig. S1). This was done by looping over the same 30 year
cycle (1901-1930) of climate forcing data from GSWP-3 during the pre-industrial period
(Table 1) and the first year (1901) of a prescribed vegetation map (ESA CCI Land Cover
Map, Bontemps et al., (2013))  –to ensure equilibrium of DOC, dissolved $CO_2$ and Active
and Slow SOC pools is driven not just by a single set of environmental factors in one year
–for a total of 400 years.  The parameter configuration adhered as close as possible to
that used in the original ORCHIDEE-MICT spinup simulations, to avoid excessive
equilibrium drift from the original SOC state (Fig. 3).
**4 Conclusion**
This first part of a two-part study has described a new branch of the high latitude
version of ORCHIDEE-MICT land surface model, in which the production, transport and
transformation of DOC and dissolved $CO_2$ in soils and along the inland water network of
explicitly-represented northern permafrost regions has been implemented for the first
time.  Novel processes with respect to ORCHIDEE-MICT include the discretisation of
litter inputs to the soil column, the production of DOC and $CO_{2(aq.)}$ from organic matter
and decomposition, respectively, transport of DOC into the river routing network and its
potential mineralisation to $CO_{2(aq.)}$ in the water column, as well as subsequent evasion
from the water surface to the atmosphere.  In addition, an improved floodplains
representation has been implemented which allows for the hydrologic cycling of DOC
and $CO_2$ in these inundated areas.  In addition to descriptions of these processes, this
paper outlines the protocols and configuration adopted for simulations using this new
model that will be used for its evaluation over the Lena river basin in the second part of
this study.
**Code and data availability**
The source code for ORCHIDEE MICT-LEAK revision 5459 is available via
http://forge.ipsl.jussieu.fr/orchidee/wiki/GroupActivities/CodeAvalaibilityPublication/
ORCHIDEE_gmd-2018-MICT-LEAK_r5459
Primary data and scripts used in the analysis and other supplementary information that
may be useful in reproducing the author's work can be obtained by contacting the
corresponding author.
This software is governed by the CeCILL license under French law and abiding by the
rules of distribution of free software. You can use, modify and/or redistribute the





software under the terms of the CeCILL license as circulated by CEA, CNRS and INRIA at
the following URL: http://www.cecill.info.
**Authors' contribution**
SB coded this model version, conducted the simulations and wrote the main body of the
paper. RL gave consistent input to the coding process and made numerous code
improvements and bug fixes.  BG advised on the inclusion of priming processes in the
model and advised on the study design and model configuration; DZ gave input on the
modelled soil carbon processes and model configuration.  MG, AT and AD contributed to
improvements in hydrological representation and floodplain forcing data. PC oversaw
all developments leading to the publication of this study. All authors contributed to
suggestions regarding the final content of the study.
**Competing interests**
The authors declare no competing financial interests.
**Acknowledgements**
Simon Bowring acknowledges funding from the European Union's Horizon 2020
research and innovation program under the Marie Sklodowska-Curie grant agreement
No. 643052, 'C-CASCADES' program.  Simon Bowring received a PhD grant. Matthieu
Guimberteau acknowledges funding from the European Research Council Synergy grant
ERC-2013-SyG-610028 IMBALANCE-P. Ronny Lauerwald acknowledges funding from
the European Union's Horizon 2020 research and innovation program under grant
agreement no.703813 for the Marie Sklodowska-Curie European Individual Fellowship
"C-Leak".

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

**Tables and Figures:**
**Table 1:** Data type, name and sources of data files used to drive the model in the study
simulations.

| Data Type | Name | Source |
|---|---|---|
| Vegetation Map | ESA CCI Land Cover Map | Bontemps et al., 2013 |
| Topographic Index | STN-30p | Vörösmarty et al., 2000 |
| Stream flow direction | STN-30p | Vörösmarty et al., 2000 |
| River surface area | | Lauerwald et al., 2015 |
| Soil texture class | | Reynolds et al. 1999 |
| Climatology | GSWP3 v0, 1 degree | http://hydro.iis.u-tokyo.ac.jp/GSWP3/ |
| Potential floodplains | Multi-source global wetland maps | Tootchi et al., 2018 |
| Poor soils | Harmonized World Soil Database map | Nachtergaele et al., 2010 |
| Spinup Soil Carbon Stock | 20ky ORCHIDEE-MICT soil carbon spinup | Based on config. in Guimberteau et al. (2018) |




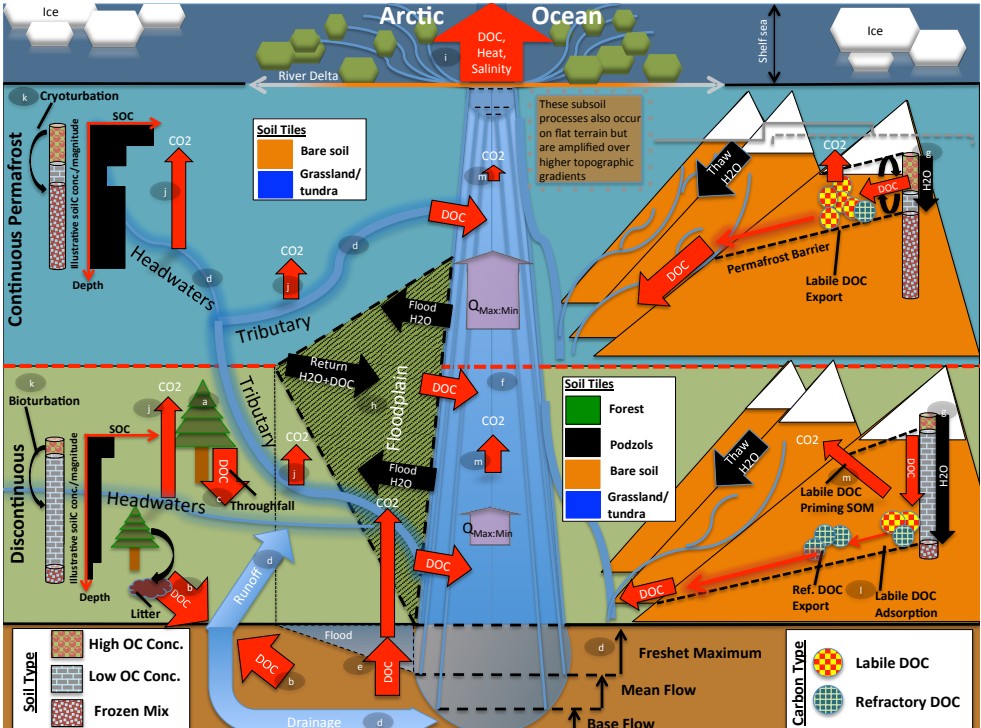

**Figure 1:** Cartoon diagram illustrating the landscape-scale emergent phenomena observed in high-latitude river systems that are captured by the processes represented in this model. Here, the terrestrial area is shown, in vertically-ascending order, as subsoil, discontinuous permafrost, continuous permafrost and the maritime boundary. Representative soil types, their distributions and carbon concentrations are shown for the two permafrost zones, as well as the different dynamics occuring on 'flat' (left) and 'sloping' land (right) arising from their permafrost designation. Carbon exports from one subsystem to another are shown in red. The relative strength of the same processes ocurring in each permafrost band are indicated by relative arrow size. Note that the high $CO_2$ evasion in headwaters versus tributaries versus mainstem is shown here. Proposed and modelled mechanisms of soil carbon priming, adsorption and rapid metabolisation are shown. The arrows $Q_{Max:Min}$ refer to the ratio of maximum to minimum discharge at a given point in the river, the ratio indicating hydrologic volatility, whose magnitude is influenced by permafrost coverage. Soil tiles, a model construct used for modulating soil permeability and implicit/explicit decomposition, are shown to indicate the potential differences in these dynamics for the relevant permafrost zones. Note that the marine shelf sea system, as shown in the uppermost rectangle, is not simulated in this model, although our outputs can be coupled for that purpose. Letter markings mark processes of carbon flux in permafrost regions and implicitly or explicitly included in the model, and can be referred to in subsections of the Methods text. These refer to: (a) Biomass generation; (b) DOC generation and leaching; (c) Throughfall and its DOC; (d) Hydrological mobilisation of soil DOC; (e) Soil flooding; (f) Landscape routing of water and carbon; (g) Infiltration and topography; (h)





Floodplain representation; (i) Oceanic outflow; (j) Dissolved carbon export and riverine
atmospheric evasion; (k) Turbation; (l) Adsorption; (m) Priming.

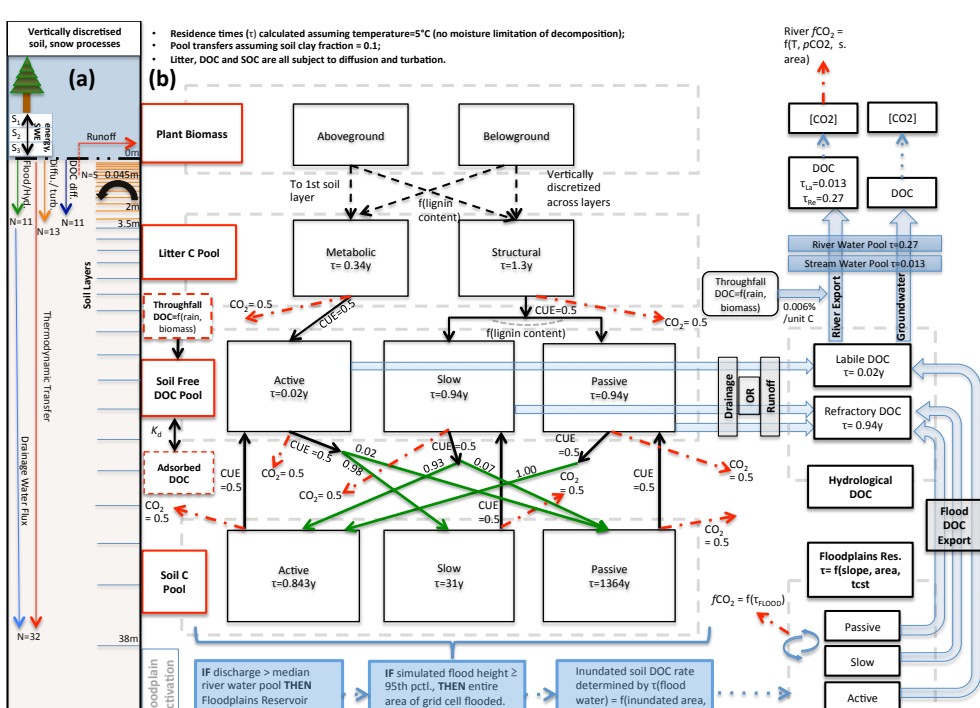

**Figure 2** :Carbon and water flux map for core DOC elements in model structure relating
to DOC transport and transformation. **(a)** Summary of the differing extent of vertical
discretisation of soil and snow for different processes calculated in the model.
Discretisation occurs along 32 layers whose thickness increases geometrically from 0-
38m. N refers to the number of layers, SWE=snow water equivalent, $S_n$ = Snow layer n.
Orange layers indicate the depth to which diffusive carbon (turbation) fluxes occur. **(b)**
Conceptual map of the production, transfer and transformation of carbon in its vertical
and lateral (i.e., hydrological) flux as calculated in the model.  Red boxes indicate meta-
reservoirs of carbon, black boxes the actual pools as they exist in the model.  Black
arrows indicate carbon fluxes between pools, dashed red arrows give carbon loss as $CO_2$,
green arrows highlight the fractional distribution of DOC to SOC (no carbon loss
incurred in this transfer), a feature of this model. For a given temperature (5°C) and soil
clay fraction, the fractional fluxes between pools are given for each flux, while residence
times for each pool ( $\tau$ ) are in each box.  The association of carbon dynamics with the
hydrological module are shown by the blue arrows.  Blue dashed boxes illustrate the
statistical sequence which activates the boolean floodplains module.  Note that for
readability, the generation and lateral flux of dissolved $CO_2$ is omitted from this diagram,
but is described at length in the Methods section.

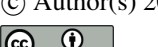




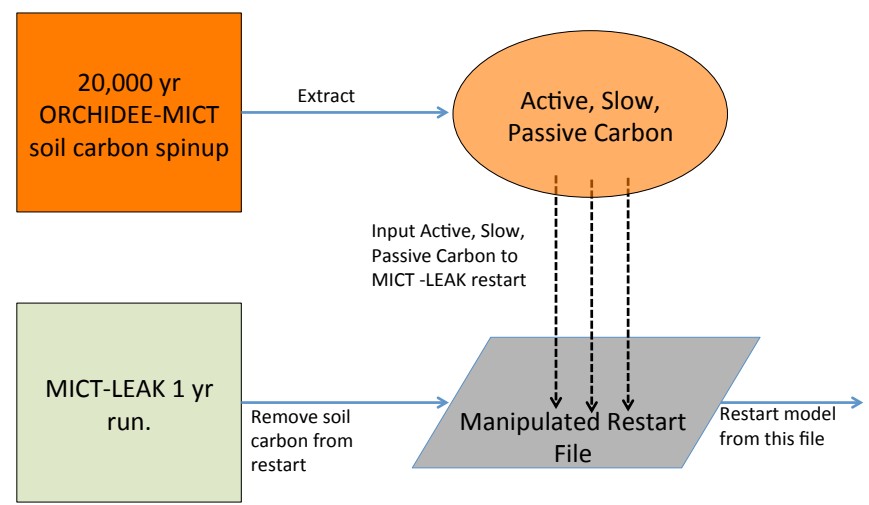

**Figure 3:** Flow diagram illustrating the step-wise stages required to implement the
model's soil carbon stock  prior to conducting transient, historical simulations.