# Peer review of "ORCHIDEE MICT-LEAK (r5459), a global model for the production, transport and transformation of dissolved organic carbon from Arctic permafrost regions, Part 1: Rationale, model description and simulation protocol."

_Geoscientific Model Development, 2018_

## Referee Comment (RC1) · Anonymous Referee #1 · 2 May 2019

The authors adapt and apply the fusion of several existing models to predict soil and riverine DOC fluxes. Part I of the study lays out the justification for this approach, as well as giving some information on the approaches and parameterization used in the model to ultimately apply the model in the Lena River Delta. While I think that this work is important and timely and I think the general approach is valid, there are several points (and inaccuracies) that need to be addressed and clarified before this manuscript can be accepted for publication. Overall, there are many parts that are quite

confusing that could be resolved by some revisions to the terminology and further edits and streamlining of the text.

Specific comments:

Line 46: the "migration of permafrost line" really only makes sense on a map. Perhaps rephrase.

line 50: the authors pulled out some very high number, I don't know where this came from. McGuire 2009 estimates a lateral flux of 80 Tg C and a net "arctic" land sink of 600-800 Tg C. That makes the DOC component $\sim$10% of NEP.

155-156: I think these numbers need to be double checked. The point of this paragraph could be clearer.

AO: This is my preference, but it wouldn't add much space to write out Arctic Ocean and it would be more intuitive to follow.

249-254: This paragraph is confusing. The points could be expanded and clarified 265-274: This is quite confusing and makes what is new here unclear.

289: This is the first mention of this site specifically, and it really comes out of nowhere. Consider introducing the site before this.

430: Typo 437: typo

444-446: Confusing. This sounds like a lake or pond

474: typo

4780480: confusing

498-490: Justification for this approach would be helpful (add supporting references)

508-525: These water pool names are really confusing.

527-534: I'm having a difficult time following this

528-540: seems like there would be less organic matter to leach from on higher slopes.

Equation 2: needs units, what does 12.011 represent? A carbon unit conversion? Equation 3, 4, 6, ditto. If these are empirically derived parameters there needs to be a reference.

Figure 1. part k. K: assumption of soil C distribution, differences between continuous and discontinuous. Don't know how well supported this is – perhaps some justification could be found in the literature.

Terminology between headwaters, tributary in figure vs. manuscript text are confusing.

―――――――――――――――――

---

## Referee Comment (RC2) · Anonymous Referee #2 · 15 May 2019

Major comments

The manuscript entitled "ORCHIDEE MICT-LEAK (r5459), a global model for the production, transport and transformation of dissolved organic carbon from Arctic permafrost regions, Part 1: Rationale, model description and simulation protocol" by Simon P.K. Bowring et al. developed a new feature, which includes the production, transport, and atmospheric release of dissolved organic carbon (DOC) from high-latitude

permafrost soils into inland waters and the ocean. Permafrost contains huge carbon deposits, and although DOC transport is one of the most important components of the current global carbon cycle, this parameter is not explicitly simulated by land surface models. The model proposed in this study is the first of its kind that directly addresses unique permafrost soil biogeochemistry and its respective processes, fully encompassing the component, on a global scale. Thus, this new feature is definitely very interesting to readers and a great advancement in global carbon cycle research. Overall, the authors need to revise the manuscript before its publication. Although there are some minor issues, I recommend that this paper be published after the suggested revisions are addressed.

My major concerns are as follows: 1. All abbreviations should be spelled out at their first usage in the Abstract as well as the main text. For instance, ORCHIDEE MICT-LEAK should be spelled out in abstract as well as the main text, where this term is first mentioned. In addition, "IPSL", "DOC-C" and "MICT" are also not spelled out. Please check for all abbreviations throughout the manuscript and define them at the first usage.

2. Line 46: "... as the permafrost line migrates poleward over time." is incorrect, because there is no line in permafrost zone. However, there is boundary between continuous and discontinuous permafrost zones, and this boundary is slowly moving poleward over time. Please correct the phrase with respect to this suggestion.

3. Please edit English grammar throughout the manuscript more carefully. For example, in line 70 "To this end" is not clear. In addition, in line 62 "metabolising" should be "metabolizing".

My minor concerns are as follows: 1. Lines 50-51: "... , the majority as dissolved organic carbon (DOC)." is not clear. Please cite some references supporting the statement. For instance, in the headwater of the Lena River basin, Suzuki et al. (2006) showed that DOC was a dominant form of riverine organic carbon transport because

inorganic carbon and particulate organic carbon (POC) transport would be negligible on the basis of their observation data. Suzuki, K. et al. (2006), Nordic Hydrology, 37(3), 303-312, doi:10.2166/nh.2006.015.

2. Line 116-117: Please consider citing Suzuki et al. (2006).

3. Line 133-134: "... , and DOC concentration are affected at watershed scale by parent material and ground ice condition (O'Donnell et al., 2016)." The statement is incomplete, because DOC concentration is also affected by active layer depth as the frozen ground table limits water infiltration into deeper soil layers, as shown by Suzuki et al. (2006).

4. Line 169: "... and greater evapotranspiration (Zhang et al., 2009)." Please consider adding the study by Suzuki et al. (2018), wherein they have shown increasing evapotranspiration from the entire Arctic circumpolar Tundra due to summer warming. Suzuki, K. et al. (2018), Remote Sensing, 10(3), 402, doi:https://doi.org/10.3390/rs10030402.

5. Line 373: " ..., non-conservative canopy DOC production rate of 9.2*10-4 g DOC-C per gram ..." is not clear. Please rewrite more clearly.

6. Line 388: "3.5 Hydrological mobilisation of soil DOC" should be "3.5 Hydrological mobilization of soil DOC".

7. Line 396: "... (see sections 'soil flooding' and 'floodplain representation')." Please add the specific section numbers.

8. Lines 520-522: Please consider citing Suzuki et al. (2006), because they observed very large DOC transport from a headwater basin of the Lena River basin.

9. Line 654: "... , such as the photochemical breakdown of riverine OC, ...". Here, OC is not clear. Please define this and add explanation.

10. For equations (1)-(6): within the equations, variables are in italics but variables in

the main text are in normal font. Please modify these for consistency.

12. In Figure 1, letters (a)-(m) are too small to read. Please enlarge the letters.

13. In the caption of Figure 1, line 1254, "(d) Hydrological mobilisation of soil DOC" should be "(d) Hydrological mobilization of soil DOC"

14. In the caption of Figure 2, line 1277 "Blue dashed boxes" should be "Blue colored boxes".

Please also note the supplement to this comment:
https://www.geosci-model-dev-discuss.net/gmd-2018-320/gmd-2018-320-RC2-supplement.pdf

———————————————————

---

## Author Comment (AC1) · 6 Jun 2019

Author Response to Interactive Comment by Anonymous Referee #1 on "ORCHIDEE MICT-LEAK(r5459), a global model for the production, transport and transformation of dissolved organic carbon from Arctic permafrost regions, Part 1: Rationale, model description and simulation protocol" by Simon P. K. Bowring et al.

Dear Anonymous Referee #1,

Thank you for taking the time to read and review our manuscript, and in doing so providing such diligent and constructive commentary for its improvement, which we hope we have been able to assimilate into its content to the greatest degree possible in our responses, which follow below.

Specific Comments

Line 46: the "migration of permafrost line" really only makes sense on a map. Perhaps rephrase. Thank you for spotting this conceptually misleading description in our text. The phrase has now been modified to "... as the boundary between discontinuous and continuous permafrost migrates poleward and toward the continental interior over time."

line 50: the authors pulled out some very high number, I don't know where this came from. McGuire 2009 estimates a lateral flux of 80 Tg C and a net "arctic" land sink of 600-800 Tg C. That makes the DOC component ∼10% of NEP.

Again, thank you for spotting this, which indeed looks misleading, and comes from taking a mix of upper and lower bounds for lateral flux and NEP, respectively. However, we can't find the 600-800TgC /yr sink you refer to in the reference cited. Referring to McGuire et al (2009) Table 2, the inversion-based terrestrial sink from Rödenbeck et al (2003) is 400 TgC/yr, that from Baker et al (2006) is 190 TgC/yr, and that from Gurney et al. (2003) is 230 TgC/yr. Because these estimates exclude the European Arctic, McGuire estimates that the 'true amount is 'less than' 0.5 PgC/yr which, given the uncertainty range from the inversion studies, means that he accepts the range of the net $CO_2$ sink as being 0-800TgC yr. In Table 6 of McGuire et al., indeed the lateral carbon flux is 39 TgC/yr excluding DIC and 83 TgC/yr with it.

In our manuscript text body, we write that " the yearly lateral flux of carbon from soils to running waters may amount to ∼40% of net ecosystem carbon exchange". This implies the total lateral carbon flux, and not the DOC. Thus, from a mid-point of 400 TgC/yr from the above-mentioned 0-800TgC/yr, we re-write the sentence as follows:

"[...] the yearly lateral flux of carbon from soils to running waters may amount to about a fifth of net ecosystem carbon exchange (∼400 TgC yr-1), about ∼40% of which may be contributed by DOC (McGuire et al., 2009). Excluding the dissolved inorganic carbon component of this flux, as well as dissolved $CO_2$ input from soils, the vast majority (85%) of riverine organic carbon discharge to the Arctic Ocean occurs as dissolved organic carbon (DOC), as described in (e.g.) Suzuki et al. (2006). "

155-156: I think these numbers need to be double checked. The point of this paragraph could be clearer.

The numbers have been double-checked and are as reported in McGuire et al. (2009), but now distinguish between total evasion and the water bodies from which these occur as follows:

" $CO_2$ evasion rates from Arctic inland waters (Fig. 1j,e,m), which include both lakes and rivers, are estimated to be 40-84 TgC yr-1 (McGuire et al., 2009), of which 15-30 TgC yr-1 or one-third of the total inland evasion flux, is thought to come from rivers. However, a recent geo-statistically determined estimate of boreal lake annual emissions alone now stands at 74-347 TgC yr-1 (Hastie et al., 2018), potentially lowering the riverine fraction of total $CO_2$ evasion. These numbers should be compared with estimates of Pan Arctic DOC discharge from rivers of 25-36 TgC yr-1(Holmes et al., 2012; Raymond et al., 2007)."

AO: This is my preference, but it wouldn't add much space to write out Arctic Ocean and it would be more intuitive to follow.

This is an understandable preference, given the already large number of acronyms contained in the document. The text has been modified accordingly.

249-254: This paragraph is confusing. The points could be expanded and clarified

Indeed, we find the same. The paragraph has been shortened and merged with the preceding paragraph. The processes that are novel are described then later in the text.

[Figure]

"However numerous improvements in code performance and process additions post-dating these publications have been included in this code. Furthermore, novel processes included in neither of these two core models are added to MICT-L, such as the diffusion of DOC through the soil column to represent its turbation and preferential stabilisation at depth in the soil, as described in Section 2.11."

265-274: This is quite confusing and makes what is new here unclear.

We have now removed the first half of this paragraph and merged the remainder with the preceding, so as to avoid unnecessary complexity and confusion. The section removed is:

" Where these differences were so large as to prove a burden in excess of the scope of this first model version, such as the inclusion of the soil carbon spinup module, they were omitted from this first revision of MICT-L. The direction of the merge –which model was the base which incorporated code from the other –was from ORCHILEAK into MICT, given that the latter contains the bulk of the fundamental (high latitude) processes necessary for this merge."

289: This is the first mention of this site specifically, and it really comes out of nowhere. Consider introducing the site before this.

We agree with this observation, and have added in the following sentence at the end of the Introduction (line 264-267).

"The choice of the Lena River basin in Eastern Siberia as the watershed of study for model evaluation owes itself to its size, the presence of floodplains and mountain areas which allow us to test the model behavior for contrasting topography, the relatively low impact of damming on the river, given that ORCHIDEE only simulates undammed fluvial 'natural flow', and its mixture of continuous and discontinuous permafrost with tundra grassland in the north and boreal forests in the south, and is described in greater detail in Part 2 of this study"

430: Typo 437: typo

Extra full-stop removed.

444-446: Confusing. This sounds like a lake or pond

The section you refer to is: "Further, in modelled frozen soils, a sharp decline in hydraulic conductivity is imposed by the physical barrier of ice filling the soil pores, which retards the flow of water to depth in the soil, imposing a cap on drainage and thus potentially increasing runoff of water laterally, across the soil surface (Gouttevin et al., 2012). In doing so, frozen soil layers overlain by liquid soil moisture will experience enhanced residence times of water in the carbon-rich upper soil layers, potentially enriching their DOC load."

This refers to the frozen vertical barrier imposed by soil freezing on hydrological transfer to deeper layers. This is why they are referred to as 'liquid soil moisture' as opposed to water body or some such, as it implies that water increases its residence time in a certain layer above the frozen portion, but does not remain static there nor 'pond' into a water body proper.

We have also added the clarification that frozen water in the form of thick ice wedges that are important for e.g. thermokarst formation, are not simulated by the present model formulation, e.g. " Note that ice wedges, an important component of permafrost landscapes and their thaw processes, are not included in the current terrestrial representation, but have been previously simulated in other models (Lee et al., 2014)".

In addition, we found some potentially misleading text in the following segment: "First, in the process of drainage DOC is able to percolate from one layer to another, through the entirety of the soil column, meaning that vertical transport is not solely determined by 11th layer concentrations, given that DOC can be continuously leached and transported over the whole soil column. "

We have adapted this section as follows: " First, as it water percolates through the

[Figure]

soil column, it carries DOC along from one layer to another through the entirety of the soil column, but this percolation is blocked when the soil is entirely frozen, i.e. it is assumed that all soil pores are filled with ice which blocks percolation. This implies that DOC transport is not just determined by what enters from the top but also by the below ground production from litter, the sorption and de-sorption to and from particulate soil organic carbon in the soil column, , its decomposition within the soil column, and water vertical transport entraining DOC between the non-frozen soil layers using the hydraulic conductivity calculated by the model as a function of soil texture, soil carbon and time-dependent soil moisture (Guimberteau et al., 2018). "

474: typo

This has been corrected.

4780480: confusing

This refers to the following section of the manuscript: "The water residence time in each reservoir depends on the nature of the reservoir (increasing residence time in the order : stream < fast < slow reservoir). More generally, residence time decreases with the steepness of topography, given by the product of a local topographic index and a constant with decreasing values for the 'slow', 'fast' and 'stream' reservoirs."

To clarify this, we have shortened and increased the conciseness of the segment as follows:

"More generally, residence time locally decreases with topographic slope and the grid-cell length, used as a proxy for the main tributary length (Ducharne et al., 2003; Guimberteau et al., 2012). This is done to reproduce the hydrological effects of geomorphological and topographic factors in Manning's equation (Manning, 1891) and determines the time that water and DOC remain in soils prior to entering the river network or groundwater."

In addition, to increase the readability of the subsection, descriptions of the hydrological module in the paragraph preceding the segment you refer to are improved upon. The original section reads: "The 'slow' water reservoir aggregates the soil drainage, i.e. the vertical outflow from the 11th layer (2 m depth) of the soil column, effectively representing 'shallow groundwater' storage. The 'fast' water reservoir aggregates surface runoff simulated in the model, effectively representing overland hydrologic flow. The 'slow' and 'fast' water reservoirs feed a delayed outflow to the 'stream' reservoir' of the adjacent subgrid-unit in the downstream direction."

The model's hydrology routing scheme is indeed a complex system, and we use the same terminology as that adopted by its architects cited to in the text, which in turn follow the terminology given to these water reservoirs in the model code.

Thus we only try to make clearer the last sentence of the paragraph with the following edit: " The 'slow' water reservoir aggregates the soil drainage, i.e. the vertical outflow from the 11th layer (2 m depth) of the soil column, effectively representing 'shallow groundwater' transport and storage. The 'fast' water reservoir aggregates surface runoff simulated in the model, effectively representing overland hydrologic flow. The 'slow' and 'fast' water reservoirs feed a delayed outflow to the 'stream' reservoir' of the next downstream sub-grid quadrant."

498-490: Justification for this approach would be helpful (add supporting references)

We assume this refers to lines 498-500 and not 498-490. This segment reads: "Active DOC flows into a Labile DOC hydrological export pool, while the Slow and Passive DOC pools flow into a Refractory DOC hydrological pool (Fig. 2b)."

This formulation follows on from prior published developments made to the model code, but is unpacked more explicitly in the section by adding the following content:

"However, because the terrestrial Slow and Passive DOC pools (Camino-Serrano et al., 2018) are given the same residence time, these two pools are merged when exported (Lauerwald et al., 2017): Active DOC flows into a Labile DOC hydrological

export pool, while the Slow and Passive DOC pools flow into a Refractory DOC hydrological pool (Fig. 2b), owing to the fact that the residence time of these latter soil DOC pools is the same in their original (ORCHIDEE-SOM) formulation (Camino-Serrano et al., 2018), and retained and merged into a single hydrological DOC pool in Lauerwald et al. (2017). The water residence times in each reservoir of each subgrid-scale quadrant determine the decomposition of DOC into $CO_2$ within water reservoirs, before non-decomposed DOC is passed on to the next reservoir in the downstream subgrid quadrant."

In addition, to improve contextual understanding, in Section 2.3 (paragraph 1) we have added the following (in red) to this section: " The non-respired half of the litter feeds into 'Active', 'Slow' and 'Passive' free DOC pools, which correspond to DOC reactivity classes in the soil column in an analogous extension to the standard CENTURY formulation (Parton et al., 1987)."

508-525: These water pool names are really confusing. 527-534: I'm having a difficult time following this Here we combine your two above comments into an adaptation to the paragraph as follows. We believe the confusion arises from our description of the fast, slow and stream reservoirs with respect to headwaters. The paragraph has been adapted as follows:

" Note that while we do not explicitly simulate headwaters as they exist in a geographically determinant way in the real world, we do simulate what happens to the water before it flows into a water body large enough to be represented in the routing scheme by the water pool called 'stream', representing a real-world river upwards of roughly stream order 4. The 'fast' reservoir is thus the runoff water flow that is destined for entering the 'stream' water reservoir, and implicitly represents headwater streams by filling the spatial and temporal niche between overland runoff and the river stem. "

528-540: seems like there would be less organic matter to leach from on higher slopes.

Yes, certainly an omission here. We have added in the line: "In addition, places with

higher elevation and slope in these regions tend to experience extreme cold, leading to lower NPP and so DOC leaching. "

Equation 2: needs units, what does 12.011 represent? A carbon unit conversion?

This has now been altered to:

"Where the pCO2 (atm.) of a given (e.g. 'stream', 'fast', 'slow' and floodplain) water pool (pCO2POOL) is given by the dissolved CO2 concentration in that pool [CO_2(aq) ], the molar weight of carbon (12.011 g mol-1) and KCO2."

Equation 3, 4, 6, ditto. If these are empirically derived parameters there needs to be a reference.

For Eq. 3 we add in the text: "Water temperature (TWATER, (°C)) isn't simulated by the model, but is estimated here from the average daily surface temperature (TGROUND, (°C)) in the model (Eq. 3), a derivation calculated for ORCHILEAK by Lauerwald et al. (2017) and retained here."

For Eq. 4, the Schmidt number that is calculated is entirely from Wanninkhof, and cited therein in the following segment: " With our water temperature estimate, both KCO2 and the Schmidt number (Sc, Eq. 4) from Wanninkhof (1992) can be calculated, allowing for simulation of actual gas exchange velocities from standard conditions.

For Eq. 6, we follow the standard CENTURY soil carbon pool formulation (Parton et al., 1987) in which rates enter black boxes of soil carbon for each grid cell and are then re-divisible over desired quantities (area/volume etc), which is why for these we did not give units, as it is simply a discrete mass over discrete time.

More specifically, the CENTURY carbon pools, rate modifiers are determined based on soil organic dynamic in Parton et al. (1987) and then evaluated on other ecosystems (Eglin et al., 2010, Dimassi et al., 2018) for ORCHIDEE. A slightly modified version of this, with the same CENTURY parameters that now account for the priming effect, was derived by Guenet et al. (2016) and included in this version. The parameters in this

equation are derived in the cited references (see Equations 1-8 in Guenet et al. 2016) and repeated in Guenet et al (2018). For clarity, we have made the following edit to the text, reflecting the fact that k is the standard decomposition rate in 1/time, the rate modifiers are zero-dimensional and SOC represents the mass of SOC, represented here by Kg as the SI unit of mass:

" Where INSOC is the carbon input to that pool, k is the SOC decomposition rate (1/dt), FOC (Kg) is a stock of matter interacting with this SOC pool to produce priming, c is a parameter controlling this interaction, SOC is the SOC reservoir (Kg), and $\theta$,$\Phi$ and $\gamma$ the zero-dimensional moisture, temperature and soil texture rate modifiers that modulate decomposition in the code, and are originally determined by the CENTURY formulation (Parton et al., 1987) and subsequently re-estimated to include priming in Guenet et al., (2016, 2018).."

Figure 1. part k. K: assumption of soil C distribution, differences between continuous and discontinuous. Don't know how well supported this is – perhaps some justification could be found in the literature.

Yes, this is only illustrative but can be found in the literature for example the top 1m of soil generally is richer in carbon in continuous over discontinuous regions, with the canonical snapshot of this captured by the NCSCD.

The caption has been edited to reflect this with the following: " (k) Turbation and soil carbon with depth (e.g. (Hugelius et al., 2013; Tarnocai et al., 2009), (Koven et al., 2015));"

Terminology between headwaters, tributary in figure vs. manuscript text are confusing.

The terminology we agree is a bit confusing because of the nomenclature that is used in the model code and in preceding papers cited herein which refer to real-world water pools like streams as 'fast reservoir' and real-world water pools like rivers as 'stream reservoir'. However, as this figure is a cartoon, we feel it appropriate to use real-world

terms for bodies such as streams and tributaries that are represented collectively in the model by both the 'fast' and 'stream' pool.

Thus in the caption text we include the following sentence: " Note that 'tributaries' in the Figure may be represented in the model by either the 'fast' or 'stream' pool, depending on their size."

Please also note the supplement to this comment:
https://www.geosci-model-dev-discuss.net/gmd-2018-320/gmd-2018-320-AC1-supplement.pdf

---

## Author Comment (AC2) · 6 Jun 2019

Author Response to Interactive Comment by Anonymous Referee #2 on "ORCHIDEE MICT-LEAK(r5459), a global model for the production, transport and transformation of dissolved organic carbon from Arctic permafrost regions, Part 1: Rationale, model description and simulation protocol" by Simon P. K. Bowring et al.

Dear Anonymous Referee #2,

[Figure]

Thank you for taking the time to read and review our manuscript, and in doing so providing such diligent and constructive commentary for its improvement, which we hope we have been able to assimilate into its content to the greatest degree possible in our responses, which follow below.

Major Comments:

1. All abbreviations should be spelled out at their first usage in the Abstract as well as the main text. For instance, ORCHIDEE MICTLEAK should be spelled out in abstract as well as the main text, where this term is first mentioned. In addition, "IPSL", "DOC-C" and "MICT" are also not spelled out. Please check for all abbreviations throughout the manuscript and define them at the first usage.

1. We have included the full expansion of the acronyms identified by your review and included them in the main body of the text. In the abstract, we have included the full spelling of 'IPSL' (Institut Pierre Simon Laplace), to reflect the fact that this may not be a well-known institute, but have decided not to do the same for 'ORCHIDEE' in the abstract, as (i) this is a relatively well-known land surface model in the modelling community, such that it may not be necessary to unpack its letters in an abstract; (ii) this unpacking is extremely lengthy, and may not be sufficiently informative to justify its inclusion to the text body of an abstract. Thus the unpacking occurs in line 72 of the text. Finally, we cannot spell out "ORCHIDEE MICT-LEAK" since the second half of the compound name (LEAK) is itself not an acronym, and refers to a version of the ORCHIDEE model called ORCHILEAK -hence our reduction of the new branch name presented in this manuscript from ORCHIDEE MICT-LEAK to ORCHIDEE M-L. The rationale for the ORCHILEAK name is now included in the text (L. 81-82) with the text " where the suffix 'LEAK' holds no acronym, and refers to the 'leakage' of carbon from terrestrial to aquatic realms). " Further, in the abstract we try to clarify the point that the presented model results from the merge of two separate code versions with the following text: " The model, ORCHIDEE MICT-LEAK, which represents the merger of previously described ORCHIDEE versions -MICT and -LEAK, mechanistically represents..."

[Figure]

2. Line 46: "... as the permafrost line migrates poleward over time." is incorrect, because there is no line in permafrost zone. However, there is boundary between continuous and discontinuous permafrost zones, and this boundary is slowly moving poleward over time. Please correct the phrase with respect to this suggestion.

2. Thank you for spotting this conceptually misleading description in our text, and for providing some helpful pointers towards its resolution. The phrase has now been modified to "... as the boundary between discontinuous and continuous permafrost migrates poleward and toward the continental interior over time."

3. Please edit English grammar throughout the manuscript more carefully. For example, in line 70 "To this end" is not clear. In addition, in line 62 "metabolising" should be "metabolizing".

3. Thank you for finding this grammatical inconsistency in our text, which reflects the inputs of authors using differing standards for English spelling. The GMD English language guidelines stipulate that ""We accept all standard varieties of English in order to retain the author's voice. However, the variety should be consistent within each article". As such, we have chosen to homogenise the text for the UK variant. Thus 'metabolize' and its variants have now all been corrected to reflect this choice of English usage in the other text (e.g. lines 125-126), as have all other verbs that contain this ('-z') difference in spelling (e.g. 'mineralization' –> 'mineralisation', line 461) throughout the text. Further, "to this end" has been changed to "for this purpose".

Minor Comments:

1. Lines 50-51: "... , the majority as dissolved organic carbon (DOC)." is not clear. Please cite some references supporting the statement. For instance, in the headwater of the Lena River basin, Suzuki et al. (2006)showed that DOC was a dominant form of riverine organic carbon transport becauseinorganic carbon and particulate organic carbon (POC) transport would be negligible on the basis of their observation data. Suzuki, K. et al. (2006), Nordic Hydrology, 37(3), 303-312, doi:10.2166/nh.2006.015.

[Figure]

Thank you for pointing out this unqualified statement. We have included the citation suggested in review.

2. Line 116-117: Please consider citing Suzuki et al. (2006).

This has been included in the text (now line 128).

3. Line 133-134: "... , and DOC concentration are affected at watershed scale by parent material and ground ice condition (O'Donnell et al., 2016)." The statement is incomplete, because DOC concentration is also affected by active layer depth as the frozen ground table limits water infiltration into deeper soil layers, as shown by Suzuki et al. (2006).

Thank you for finding this error in conceptualisation. Indeed, we agree with the reviewer that this is a critical determinant of DOC conentrations, and have altered the text to reflect this with "DOC concentrations are affected at watershed scale by parent material, ground ice content (O'Donnell et al., 2016) and active layer depth (Suzuki et al., 2006). "

4. Line 169: "... and greater evapotranspiration (Zhang et al., 2009)." Please consider adding the study by Suzuki et al. (2018), wherein they have shown increasing evapotranspiration from the entire Arctic circumpolar Tundra due to summer warming. Suzuki, K. et al. (2018), Remote Sensing, 10(3), 402, doi:https://doi.org/10.3390/rs10030402. Thank you for alerting us this additional citation that further strengthens the assertions made in this portion of the text (now line 187).

5. Line 373: " ..., non-conservative canopy DOC production rate of 9.2*10-4 g DOC-C per gram ..." is not clear. Please rewrite more clearly.

Indeed, on reflection, this sentence is not particularly straightforward and has been adapted to make what has been calculated clearer to the reader. It now reads " From this we obtain a constant tree canopy DOC production rate of 9.2*10-4 g DOC-C per

gram of leaf biomass per day (Eq. 1). This is the same for all PFTs except those representing crops, for which this value equals 0, reflecting how at a very general level, crops are small and tend no to be characterised by high organic acid loss rates from leaves due to e.g. aphids, due to human control." (now lines 394-399).

6. Line 388: "3.5 Hydrological mobilisation of soil DOC" should be "3.5 Hydrological mobilization of soil DOC".

This has now been included (see Major Comments Response (3)).

7. Line 396: "... (see sections 'soil flooding' and 'floodplain representation')." Please add the specific section numbers.

Here we realise that the section headings had changed since this part was written, and we had since merged the segments discussing floodplain representation. This is now reflected in the text body (line 424) which now reads: "(see section 2.8, 'Representation of floodplain hydrology and their DOC budget')."

8. Lines 520-522: Please consider citing Suzuki et al. (2006), because they observed very large DOC transport from a headwater basin of the Lena River basin.

Thank you for your suggestion. This has now been included.

9. Line 654: "... , such as the photochemical breakdown of riverine OC, ...". Here, OC is not clear. Please define this and add explanation.

Thank you, this has been corrected to "dissolved organic carbon" (now line 691).

10. For equations (1)-(6): within the equations, variables are in italics but variables in the main text are in normal font. Please modify these for consistency.

Indeed, we had not noticed this inconsistency in the text, which has now been edited accordingly throughout.

12. In Figure 1, letters (a)-(m) are too small to read. Please enlarge the letters.

(note, no 11. in the original review document). The font size for the letter subheadings has been increased from 8 point to 12 point in Figure 1.

13. In the caption of Figure 1, line 1254, "(d) Hydrological mobilisation of soil DOC" should be "(d) Hydrological mobilization of soil DOC"

This remains as was (see choice of English in Major Comments (3)).

14. In the caption of Figure 2, line 1277 "Blue dashed boxes" should be "Blue colored boxes".

This change has been included in the document.

Please also note the supplement to this comment:
https://www.geosci-model-dev-discuss.net/gmd-2018-320/gmd-2018-320-AC2-supplement.pdf